# Electroless Ni-P-MoS_2_-Al_2_O_3_ Composite Coating with Hard and Self-Lubricating Properties

**DOI:** 10.3390/ma15196806

**Published:** 2022-09-30

**Authors:** Shalini Mohanty, Naghma Jamal, Alok Kumar Das, Konda Gokuldoss Prashanth

**Affiliations:** 1Department of Mechanical Engineering, Indian Institute of Technology (ISM), Dhanbad 826004, India; 2Department of Mechanical and Industrial Engineering, Tallinn University of Technology, 12616 Tallinn, Estonia; 3Erich Schmid Institute of Materials Science, Austrian Academy of Sciences, Jahnstrasse 12, A-8700 Leoben, Austria; 4Centre for Biomaterials, Cellular and Molecular Theranostics, School of Mechanical Engineering, Vellore Institute of Technology, Vellore 630014, India

**Keywords:** electroless plating, composite, wettability, microhardness, lubricant, coating

## Abstract

The work aimed to produce Ni-P-MoS_2_-Al_2_O_3_ on Al-7075 alloys with multiple attributes through an electroless (EL) plating route. The effects of additives (MoS_2_ and Al_2_O_3_) in the EL bath on the surface morphology, topography, hardness, composition (phase and elemental), roughness, wettability, and coating thickness were evaluated. Results indicate a substantial enhancement in microhardness of the EL-coated surfaces by 70% (maximum hardness = ~316 HV) using powders, and 30% (244 HV) without powders. The maximum coating thickness and water contact angle obtained with powders were 6.16 μm and 100.46°, respectively. The coefficient of friction for the samples prepared using powders was 0.12, and for the base material it was 0.18. The compositional analysis through EDS and XRD suggested the incorporation of a hard and lubricious layer on the EL-coated surface owing to the presence of different phases of Al, Mo, P, Zn, O, and S. Therefore, the resulting coating surfaces impart hardness, self-lubrication, hydrophobicity, and wear resistance simultaneously.

## 1. Introduction

Al and Al-based alloys, owing to their inherent light weight, toughness, and high strength, have gained attention as important engineering materials for aerospace industries [1,2,3]. Of all the available alloys, Al-7075 occupies a vital place in the designing and manufacturing of aerospace components [4,5,6]. However, they are prone to high-stress corrosion cracking that fails in high-stress concentration regions, such as riveted zones in airplanes [7,8]. Therefore, improving the performance of the Al-7075 surface in terms of mechanical, corrosion, and tribological properties is essential. These desirable properties can be achieved at low deposition temperatures using the electroless (EL) Ni-P plating route, which is compatible with heat-sensitive alloys [9]. Ni-P-based electroless (EL) deposition is produced by auto-catalytic chemical reduction of Ni-cations obtained from the liquid solution of metallic salt using reducing agents [10]. The reducing agent is usually sodium hypophosphite (in this case: Ni-P) or sodium borohydride (in this case: Ni-B), depending upon the substrate composition [11]. EL deposition can process both electrically conductive and electrically non-conductive materials [12]. The alloy’s chemical content strongly affects the substrate properties and controls the bath temperature, pH, and Ni concentration [13].

EL-Ni-P coatings find their applications in the coating of corrosion-resistant components used in extreme environmental conditions in place of sacrificial layers [14,15,16]. In the past, several reports have dealt with the tribo-mechanical and corrosive behavior of Ni-P coatings by the EL method [17]. Recently, the distribution of P and a surface morphological study of the coated surface have been addressed in detail [18]. Typically, the corrosion resistance of the EL-plated components depends on the degree of crystallinity, P content, phase change, coating thickness, porosity on the surface, size, and orientation of grains [19,20,21]. As per a study by Gutierrez et al., the corrosion resistance of the Ni-P EL-coated component depends upon the P content, the extent of internal stress, and the amorphous state [22]. The performance evaluation of Ni-P coated surfaces has been conducted in the context of the factors mentioned above in several studies [8,9,10]. Vlaic et al. demonstrated that amorphous coatings possess high corrosion resistance and low hardness due to surface flaws and the absence of grain boundaries [23]. Aiming to achieve multiple good attributes in a single component for diverse applications, composite coating through the EL deposition route has come into existence [24,25,26]. The EL bath is modified using powders such as metallic carbides, nitrides, oxides, solid lubricants, and PTFE, to improve the surface properties [27,28]. Uysal developed a layer of Ni-P-TiO_2_-GO using oxides of both graphene and Ti particles suspended in an EL bath, to which he attributed improved wear and corrosion properties [29]. Hu et al. generated Ni-P-Al_2_O_3_ coatings over Mg-alloys, thereby improving their corrosion resistance and hardness [30]. Taye et al. used WS_2_ and Al_2_O_3_ in an EL bath to form a Ni-Al_2_O_3_-WS_2_ composite on an Al substrate [27]. They confirmed an upsurge in microhardness values of the Al substrate, and the maximum coating thickness was obtained with 3 g/L Al_2_O_3_ and 1.4 g/L WS_2_, respectively [27]. In a similar study by He et al., Ni-P-MoS_2_ was formed on a mild steel substrate, imparting a lower coefficient of friction than the pure Ni-P coating [31].

Although several studies have been conducted on EL plating with/without powders (as reinforcement), limited research has explored forming both hard and lubricating surfaces. In the present study, aerospace-grade Al alloy (Al7075) was coated through the EL route, imparting hard, self-lubricating, and hydrophobic properties. An extensive analysis was performed of the surface morphology, topography, hardness, composition, roughness, wettability, and coating thickness for various reinforcement ratios (MoS_2_ and Al_2_O_3_).

## 2. Materials and Methods

### 2.1. Material

Al-7075 alloys of dimensions 12 × 12 × 12 mm^3^, cut by a wire-cut electrical discharge machine, were utilized to carry out the EL deposition. The detailed composition of the alloy is presented in Table 1. To obtain a contaminant-free surface, the samples were polished and cleaned in deionized water, and subsequently with acetone.

### 2.2. Experimental Procedure

The experimental set-up (Figure 1a) constitutes a magnetic stirrer with a supporting stand to hang the substrate. The samples were dipped into an EL bath and placed over the stirrer. A thermometer-cum-pH sensor was affixed to the frame/stand to control the necessary conditions.

As Al-based coatings are prone to oxidation when exposed to air/water, it is difficult to achieve good coating adhesion because the oxide layer acts as a barrier between the coating and substrate. Thus, pretreatment in the form of double zincating (ZnO + aqueous NaOH) was performed on the Al substrate before the EL-deposition process [32]. The flow chart for the double-zincating process (desmutting, etching, and zincating) is shown in Figure 1b. A desmutting solution comprising 50:50 concentrations of HNO_3_ (69% purity) and deionized (DI) water, was prepared at room temperature (300 K). The etching solution was made by dispersing NaOH in DI water. The zincate layer comprised ZnO, NaOH pellets, and FeCl_3_ at the ratio shown in Table 2. Post-zincating, the samples were immersed straight away into the Ni-P EL bath with a certain composition, as shown in Table 3.

Throughout the experiments, the pH and temperature were maintained between 4 and 4.8 and 70 and 80 °C, respectively. The temperature should not exceed 90 °C, as it would then be challenging to maintain the pH and may lead to total decomposition of the bath [7]. The magnetic stirring limit was fixed to 600 rpm, and the deposition process was accomplished in 30 min. A total of 10 experiments (with three repetitions) were performed at different concentrations of Al_2_O_3_ (2, 3, and 4 g/L; average particle size = 12 μm) and MoS_2_ (0.5, 1, 1.5, and 2 g/L; average particle size = 7 μm) in the EL bath. The scanning electron microscopy images and phase analysis of the powders are shown in Figure 2. The Al_2_O_3_ powders were present in the α-phase (corundum) of the hexagonal structure. The coated specimen was then taken out and cleaned with an ultrasonicated acetone-filled bath to remove any adhered particles on the surfaces.

### 2.3. Material Characterization

Detailed characterization was conducted to study the coated surfaces’ mechanical and metallurgical properties. The coating thickness was measured with Field Emission Scanning Electron Microscopy (FESEM, make: Supra 55; from Carl Zeiss, Oberkochen, Germany), along the transverse direction of the coated samples, which were diamond polished along the cross-section. An average of four readings were considered for further analysis. The top surface morphology of each coated sample was also observed using FESEM. A Vickers micro-hardness tester (model: Economet VH-1MD, Chennai Metco, Chennai, India) was used to estimate the extent of the hardness of each composite coating at a constant load of 200 gf with a 10 s dwell time. Measurements were taken at three different points, and the mean values are used for further evaluation. The average surface roughness parameter was estimated using a probe-type surface roughness tester (MITUTOYO-SJ-210, Kanagawa, Japan, 0.3 mm cut-off length, 4 μm tip diameter, 4 mN measurement force) at three different sites. An Atomic Force Microscope (AFM, model: Nanoscope V, Bruker Corporation, Bremen, Germany) was used to examine the surface topography of the coated surface. Readings (scan area: 50 µm × 50 µm) were taken under ambient atmospheric conditions. A Berkovich-type diamond tip cantilever with 260 μN constant force was used for the measurements. The energy dispersive spectrum (EDS) was used to detect the elemental composition of each coating’s surface. X-ray diffraction analysis (XRD, Model: D2 PHASER, Bruker Corporation, Bremen, Germany) was used to evaluate the phase formation in the coated substrates. Micro-scratch tests (Universal tribometer, MFT 500, Rtech Instruments, San Jose, CA, USA) were conducted at a constant load of 5 N for 15 s, covering a unidirectional scratch length of 4 mm to study the friction characteristics of the coatings.

## 3. Results and Discussion

### 3.1. Coating Thickness Measurement

The coating thickness measurements were taken at four different sites (Figure 3), and the average was recorded, as shown in Table 4. The thicknesses of deposits for different powder contents are shown in Figure 3a–g, where the substrate and coating can be easily distinguished. Figure 3h depicts the variation of coating thickness with respect to the powder content in the EL bath. It can be observed in Figure 3h that when the MoS_2_ powder increased from 0.5 to 1 g/L at a constant value of Al_2_O_3_, coating thickness decreased drastically, but with further addition of MoS_2_ powder to the EL bath, i.e., at 1.5 and 2 g/L, coating thickness continued to increase. An exciting trend can also be observed in Figure 3h, wherein for constant MoS_2_ content in the bath, the composite coating thickness first increases and then decreases with an increase in the Al_2_O_3_ content in the bath. It can also be observed that with 2 g/L of Al_2_O_3_ powder content, the thickness ranged between 4.20 and 4.49 µm, and on increasing the Al_2_O_3_ content to 3 g/L in the bath, the coating thickness increased in the range of 4.75–6.16 µm. An abrupt decrease in coating thickness was observed on further increasing the Al_2_O_3_ content to 4 g/L (Figure 3h). Figure 4c shows the coating surface morphology with Al_2_O_3_ 3 g/L and MoS_2_ 0.5 g/L. There was a homogeneous deposition of Al_2_O_3_ and MoS_2_ powder in the Ni matrix. This may have been the reason behind the maximum coating thickness. Moreover, Al_2_O_3′_s abrasive nature creates a hindrance during deposition. A larger Al_2_O_3_ (4 g/L) concentration may create random movement in the EL bath, suppressing the deposition process [32]. Figure 4g depicts the morphology of the surface coating with Al_2_O_3_ 4 g/L and MoS_2_ 1 g/L powder content. It clearly displays the heterogeneity in the powder distribution, which may lead to poor deposition and hence the lowest coating thickness.

### 3.2. Microhardness

Figure 5f depicts the variation in the average microhardness of the composite coating with powder concentrations. It can be observed that microhardness increased with the upsurge in Al_2_O_3_ content in the EL bath at constant MoS_2_ (0.5 g/L). It was due to the presence of Al_2_O_3_ particles, which enhanced the phase structure of coated surfaces, thereby improving their micro-hardness values. However, for constant Al_2_O_3_ content, an increase in MoS_2_ content led to a continuous decrease in hardness. MoS_2_ is a solid lubricating material, softer than nickel, and its existence in the scene compelled a reduction in the crystal size of the Ni-MoS_2_ coating. The rise in grain boundaries because of the change in the crystal size may hinder dislocation mobility [33].

The maximum microhardness obtained for coatings was ~316 HV, whereas for the Al alloy, the microhardness value was only ~180 HV. Hence, the microhardness of the composite coatings increased by ~76% compared to the substrate, and the microhardness of Ni-P coating increased only by 30% (~244 HV). The maximum value of microhardness was obtained with 4 g/L Al_2_O_3_ and 0.5 g/L MoS_2_ powder content and the minimum value with 4 g/L Al_2_O_3_ and 2 g/L MoS_2_ powder content. This suggests that the Al_2_O_3_ concentration plays a vital role in imparting hardness and MoS_2_ towards lowering micro-hardness.

### 3.3. Average Surface Roughness Parameter

Figure 5c shows the trend of average surface roughness (R_a_) with varying powder concentrations in the EL bath. R_a_ varied in the range of 0.45 to 0.85 µm, which is less than the results published by He et al. [34], i.e., greater than 2 µm while using MoS_2_ alone. Figure 5c shows that for a particular concentration of Al_2_O_3_, R_a_ first increased and then decreased with increasing MoS_2_ concentration. However, as the concentration of Al_2_O_3_ increased from 2 to 3 g/L, R_a_ increased abruptly. This may have been due to the agglomeration of Al_2_O_3_ particles on the coated surface and the generation of larger peaks, thereby increasing the R_a_ values. Further addition of Al_2_O_3_ to the EL bath decreased the R_a_ value, which may have been due to the filling of micro-gaps on the Ni-P layer, thereby enhancing the surface quality [35]. The highest average surface roughness (0.85 µm) was obtained with 3 g/L Al_2_O_3_- and 1 g/L MoS_2_-coated samples. A higher range of R_a_ values was obtained with 3 g/L Al_2_O_3_, and the same for coating thickness. The entrapment of powders in the micro-gaps on the coated surfaces resulted in a surprisingly high deposition rate, thereby increasing the coating thickness. Thus, the coated samples’ morphology (Figure 6c) showed overlapped cauliflower-like depositions.

### 3.4. Topological Study through Atomic Force Microscopy

The topographical analysis was carried out for the samples with 4 g/L Al_2_O_3_ and 2 g/L MoS_2_ (Figure 5a,b) and 3 g/L Al_2_O_3_ and 1 g/L MoS_2_ (Figure 5c,d) powder content. Figure 5 shows the 2D and 3D topography of the coatings. The variations in color in the image indicate the heights of peaks and valleys. The dark brown areas indicate valleys, and the whitish areas show peaks on the surface. Furthermore, with the increase in MoS_2_ concentration in the EL bath, i.e., from 1 g/L (Figure 5c) to 2 g/L (Figure 5a), the intensity of peaks and valleys decreased. This may have been due to the lubricating effect of MoS_2_ that allows the formation of a smoother surface in the former case.

### 3.5. Morphological Analysis

Surface morphology was examined through secondary electron (SE) and backscattered electron (BSE) images captured through FESEM. Figure 6 depicts the SE images of the surfaces. Figure 6a demonstrates the surface morphology of the coatings, wherein a bubble-like structure indicates the presence of the Ni-P phase. Figure 6b–d show the morphologies of the composite coated with 0.5 g/L MoS_2_ and 2 g/L Al_2_O_3_; 2 g/L MoS_2_ and 3 g/L Al_2_O_3_; or 2 g/L MoS_2_ and 4 g/L Al_2_O_3_ powder content. In Figure 6c, the accumulation of Al_2_O_3_ can be seen as white patches. The cauliflower-like surfaces mark the appearance of MoS_2_ particles. The coating surface has some micro-cracks and pores as coating defects. The deposition process begins preferentially on scratches and imperfections on the substrate’s surface [28]. With continuous exposure of powders in the EL bath to the substrate, there is the formation of nodular structures as seen in the coatings, which accounts for homogeneity, when prepared with 3 g/L Al_2_O_3_ and 0.5 g/L MoS_2_ (Figure 6c). On the contrary, the heterogeneity may be due to the uneven distribution of powder particles in the EL bath. This also indicates the presence of Al_2_O_3_-MoS_2_ co-deposition, an Ni-P phase, and micropores. Thus, it can be concluded that there was successful co-deposition of powder particles.

### 3.6. Compositional Analysis

#### 3.6.1. Energy Dispersive Spectroscopy Analysis

Figure 7a shows the EDS plot of the EL-Ni-P coating, indicating the presence of Ni and P elements. Further, the EDS plot of the sample with 2 g/L Al_2_O_3_ and 2 g/L MoS_2_ powder content is shown in Figure 7b, indicating the appearance of O, Al, S, Mo, Ni, P, and Zn elements, thus confirming the formation of the composite coating.

#### 3.6.2. Elemental Mapping through Energy Dispersive Spectroscopy

To verify the results obtained from EDS, mapping of elemental composition was conducted. Figure 8a illustrates the spectrum area derived for mapping. In Figure 8b, orange represents the presence of Al; in Figure 8c, the green represents the presence of Ni; purple represents Mo (Figure 8d); the red represents S (Figure 8e); yellow represents P (Figure 8f); cyan represents oxygen (Figure 8g); and blue represents Zn (Figure 8h). The result validates the successful deposition of the composite coating elements.

#### 3.6.3. X-ray Diffraction Analysis

XRD was employed to ascertain the phases formed in the coating material. The raw data were fed to software (X’pert High Score Plus) to detect the phases and diffraction peaks. Additionally, different crystalline phases of elements and compounds were also identified. The comparative XRD plots of the Ni-P coating and composite coating with 3 g/L Al_2_O_3_ and 1 g/L MoS_2_ powder content are shown in Figure 9. Figure 9a reveals the presence of different phases, including Al, P, Al-Ni, NiP_2_, MoS_2_, and Ni-Fe. The presence of Ni, NiP_2_, Ni-Fe, and Al-Ni peaks confirms Ni ions’ transfer to the substrate. Figure 9b shows the diffraction peaks of the composite coating constituting Al, Al-Ni, MoS_2_, and Ni-Fe. The peaks of MoS_2_ demonstrate the improvement in the lubricity properties of the samples [36,37].

### 3.7. Wettability Studies

The wettability of the prepared coatings was compared with Al-7075 using water (distilled) contact angle (θ_c_). It was measured using an in-house-built laboratory-scale set-up (Figure 10a), constituting a needle-syringe arrangement (placed at 10 mm height from the surface), a high-speed camera, and an incandescent light source. The tests were performed at room temperature (300 K), and the camera captured images of the drops touching the surface [38]. The images were subjected to Image-J software to estimate the water contact angles. For further analysis, the average (of three readings) θ_c_ was considered for each sample taken up for the study. If θ_c_ ≥ 90°, the surface is hydrophobic; θ_c_ ≤ 90° for a hydrophilic surface. Five samples with different powder contents were selected for estimating the wettability; we plotted the values in Figure 10b. It is evident from the plot that θ_c_ for the substrate was ~60°, but for coated samples varied between 78.7° and 102.4°. Without the use of powders in the EL bath, θ_c_ = 78.7°. At a constant MoS_2_ concentration (0.5 g/L) in the EL bath, θ_c_ varied between 96.3° (4 g/L Al_2_O_3_) and 102.4° (2 g/L Al_2_O_3_), which accounts for the formation of hydrophobic surfaces. Similarly, at maximum MoS_2_ content (4 g/L), θ_c_ varied between 91° (2 g/L Al_2_O_3_) and 100° (4 g/L Al_2_O_3_). This indicates that the cauliflower-like shapes formed over the coated surfaces (as discussed in Section 3.5) partially restricted the direct water droplet contact with the composite layer. Due to the deposition phenomenon, hierarchical structures were formed on the substrate due to sub-millimeter-sized surface roughness [31]. These allowed air-entrapment within the hierarchical surface roughness, thereby enhancing the hydrophobicity. This property of the coated surfaces forbids contact between the corrosive fluids or water in the working environment with the substrate, thereby protecting the component [39].

### 3.8. Scratch Test

Scratch tests were performed on the coated samples to assess the wear characteristics of the surfaces. Two differently coated samples, i.e., with powder (sample 10) and without powder (sample 1), were taken up for the unidirectional micro-scratch tests, and the results were compared with that of the substrate. Figure 10c shows the coefficient of friction (COF) versus time graphs for different coated samples. The COF of the coated samples prepared without the powders was 0.13, that for samples coated using powders was 0.12, and that for the substrate was 0.18. The sample coated with 4g/L Al_2_O_3_ and 0.5 MoS_2_ (sample 10) possessed the least COF and maximum microhardness (315.83 HV). Wear resistance is proportional to hardness, and it may be affirmed that the addition of powders enhances the wear-resistant properties on the coated surfaces with respect to the uncoated specimen.

## 4. Conclusions

Ni-P-MoS_2_-Al_2_O_3_ composite coatings were produced on Al-7075 using an EL-deposition process. The effects of different powder ratios (MoS_2_ and Al_2_O_3_) were studied, and the following observations were made:The coating thickness for monolithic Ni-P plating was 8.23 μm, and it varied between 1.11 μm (4 g/L Al_2_O_3_ and 1 g/L MoS_2_) and 6.16 μm (3 g/L Al_2_O_3_ and 0.5 g/L MoS_2_) for powders mixed in the EL bath.The micro-hardness for Ni-P deposited surface was 244.10 HV (30% increment), and that of the substrate was 179.68 HV. It significantly increased by up to 70% for composite coated samples (222.95 to 315.83 HV).EDS analysis affirmed Ni, P. Al, Mo, S, Zn, and O as elements on the coated substrate.The XRD study indicated the presence of P and Al on the coated surfaces and some intermetallic compounds, such as Ni-Fe, Al-Ni, NiP_2_, and MoS_2_.The recorded water contact angle (θ_c_) for the substrate was ~60°, and for the coated samples varied between 78.7° (without powders) and 102.4° (2 g/L Al_2_O_3_ and 0.5 g/L MoS_2_). Without powders in the EL bath, the θ_c_ was 80.7°.

The composite coatings impart exclusive properties, such as improved hardness, hydrophobicity, and self-lubrication. Although the work presented a brief study by varying the powder ratios, an extensive study on the P content, corrosion, and adhesion is intended for the future.

## Figures and Tables

**Figure 1 materials-15-06806-f001:**
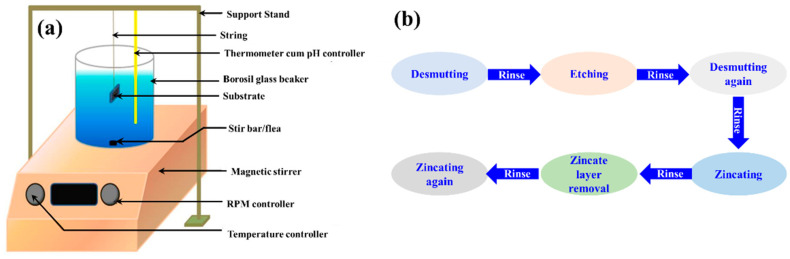
Schematic diagram illustrating the (**a**) experimental set-up and (**b**) the procedure for the electroless deposition process.

**Figure 2 materials-15-06806-f002:**
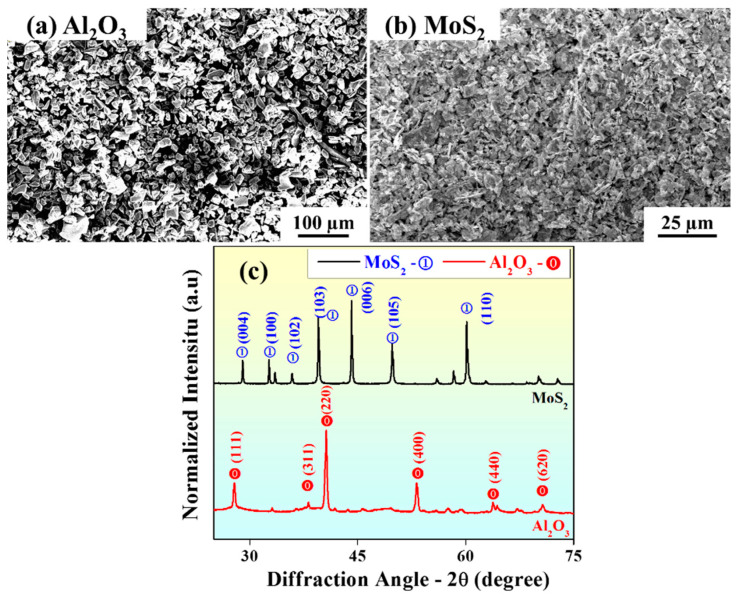
Scanning electron microscopy images showing the powder particle size distributions of (**a**) Al_2_O_3_ and (**b**) MoS_2_, and (**c**) their corresponding X-ray diffraction patterns.

**Figure 3 materials-15-06806-f003:**
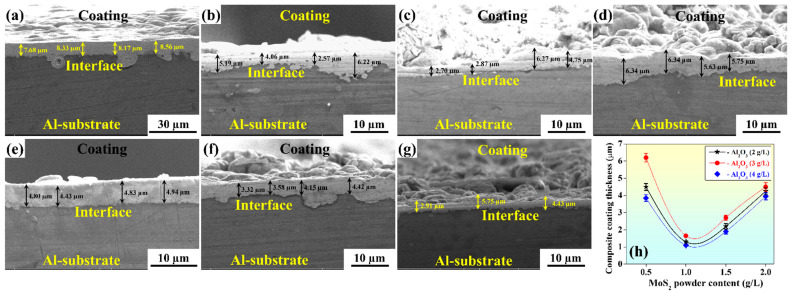
Composite coating thickness for (**a**) Ni-P coating without powder and 2 g/L Al_2_O_3_ with (**b**) 0.5 g/L MoS_2_ or (**c**) 2 g/L MoS_2_ powder content; 3 g/L Al_2_O_3_ with (**d**) 0.5 g/L MoS_2_ or (**e**) 2 g/L MoS_2_; 4 g/L Al_2_O_3_ with (**f**) 0.5 g/L MoS_2_ or (**g**) 2 g/L MoS_2_; and (**h**) variation of coating thickness with different powder concentrations.

**Figure 4 materials-15-06806-f004:**
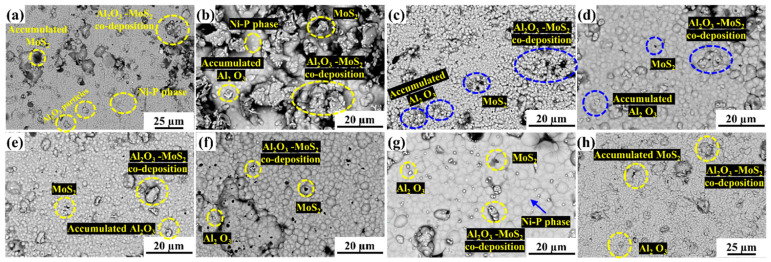
Back-scattered electron microscopy images of composite coating surfaces with (**a**) Al_2_O_3_ 2 g/L, MoS_2_ 1 g/L; (**b**) Al_2_O_3_ 2 g/L, MoS_2_ 2 g/L; (**c**) Al_2_O_3_ 3 g/L, MoS_2_ 0.5 g/L; (**d**) Al_2_O_3_ 3 g/L, MoS_2_ 1 g/L; (**e**) Al_2_O_3_ 3 g/L, MoS_2_ 1.5 g/L; (**f**) Al_2_O_3_ 4 g/L, MoS_2_ 0.5 g/L; (**g**) Al_2_O_3_ 4 g/L, MoS_2_ 1g/L; and (**h**) Al_2_O_3_ 4 g/L, MoS_2_ 1.5 g/L powder content.

**Figure 5 materials-15-06806-f005:**
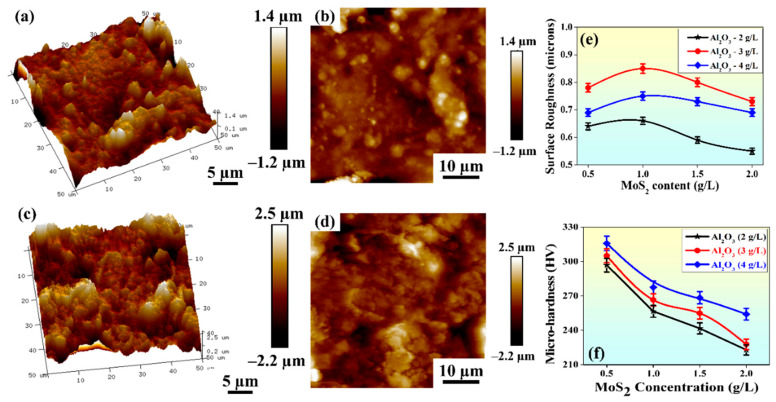
Atomic force microscopy images of the coated surface with (**a**,**b**) 4 g/L Al_2_O_3_ and 2 g/L MoS_2_ and (**c**,**d**) 3 g/L Al_2_O_3_ and 1 g/L MoS2 powder content; variation of (**e**) average surface roughness; and (**f**) variation of micro-hardness with powder concentration.

**Figure 6 materials-15-06806-f006:**
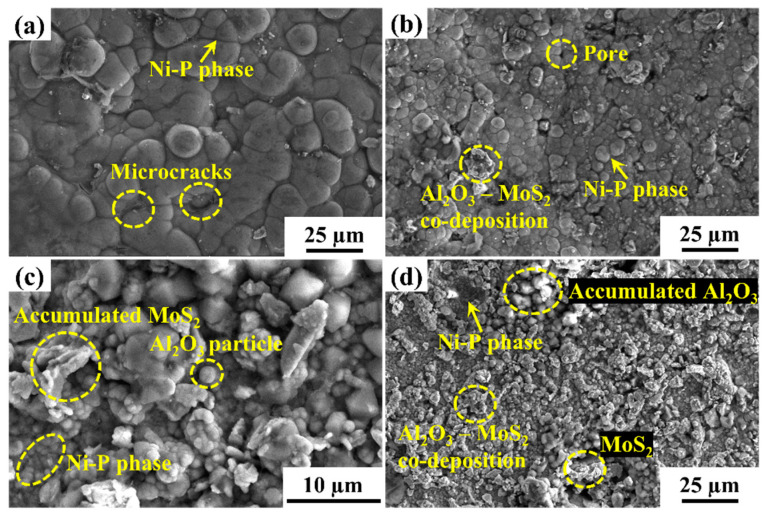
SE images of (**a**) an Ni-P coating without powder; composite coatings with (**b**) 0.5 g/L MoS_2_ and 2 g/L Al_2_O_3_, (**c**) 2 g/L MoS_2_ and 3 g/L Al_2_O_3_, and (**d**) 2 g/L MoS_2_ and 4 g/L Al_2_O_3_.

**Figure 7 materials-15-06806-f007:**
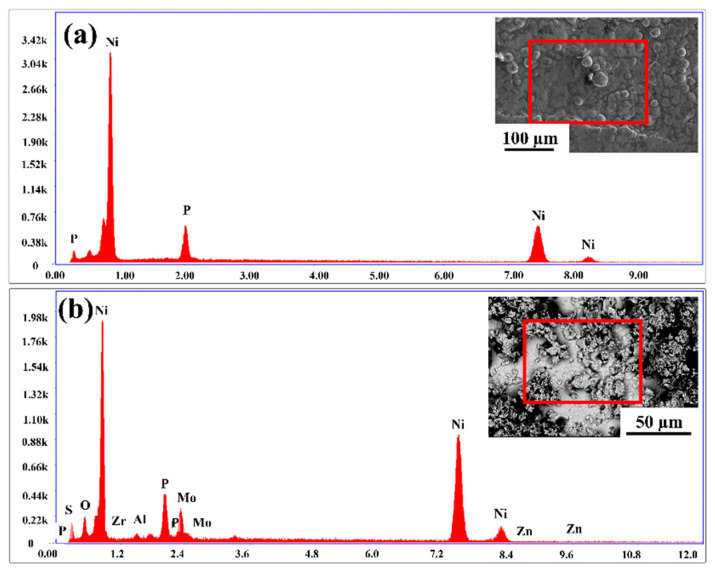
The elemental concentrations (EDS) of (**a**) the Ni-P coating and (**b**) composite coating with 2 g/L Al_2_O_3_ and 2 g/L MoS_2_ powder content.

**Figure 8 materials-15-06806-f008:**
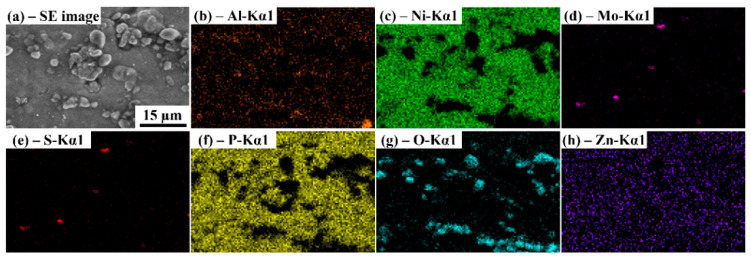
Energy dispersive spectroscopy mapping of the composite coating surface with 3 g/L Al_2_O_3_ and 1 g/L MoS_2_ powder concentration indicating (**a**) SE image, elements (**b**) Al, (**c**) Ni, (**d**) Mo, (**e**) S, (**f**) P, (**g**) O, and (**h**) Zn.

**Figure 9 materials-15-06806-f009:**
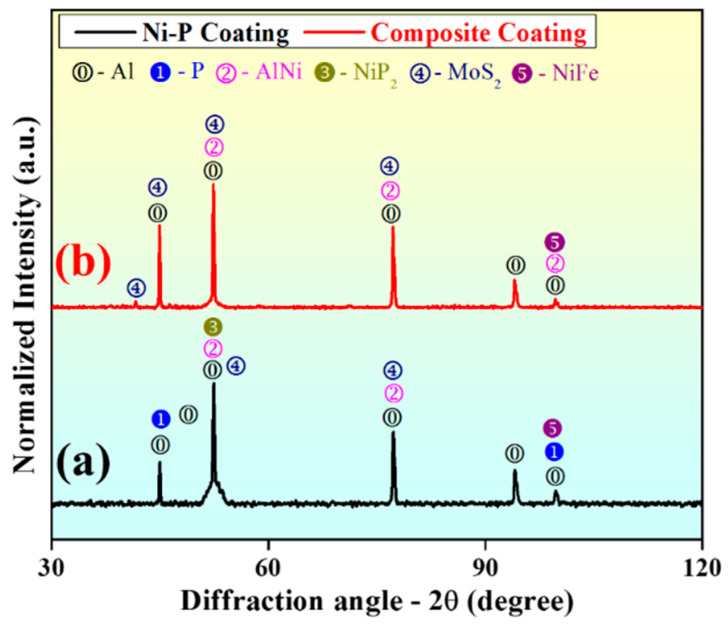
XRD plots of (**a**) Ni-P coating and (**b**) composite coating with 3g/L Al_2_O_3_ and 1 g/L MoS_2_ powder content.

**Figure 10 materials-15-06806-f010:**
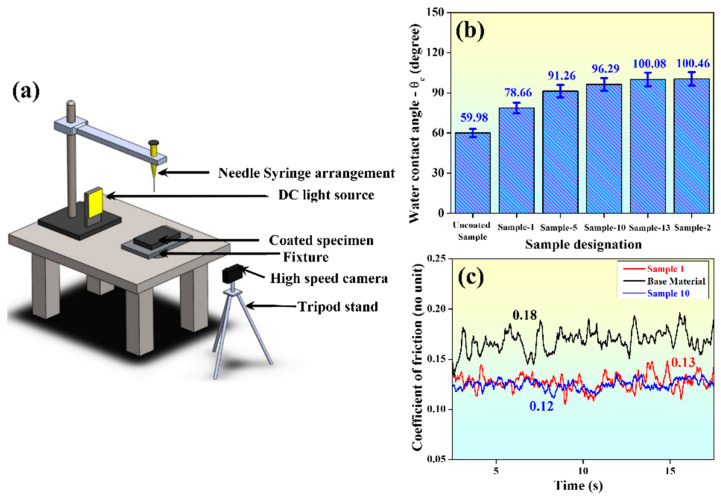
(**a**) In-house-built water contact angle measurement set-up [36], (**b**) water contact angles for different samples, and (**c**) variation of the coefficient of friction over time during the micro-scratch test.

**Table 1 materials-15-06806-t001:** Chemical composition of the substrate (Al7075) used for the EL-deposition/coating process.

Elements	Mg	Zn	Fe	Cu	Mn	Si	Cr	Ti	Al
wt. %	2.50	5.50	0.50	1.60	0.40	0.40	0.15	0.20	88.95

**Table 2 materials-15-06806-t002:** Chemical composition of the zincate bath used on the Al substrate before the EL-deposition process.

Composition	Concentration (g/L)
FeCl_3_	22.50
ZnO	60
NaOH	56

**Table 3 materials-15-06806-t003:** Electroless bath content used for the deposition process.

Component	Concentration (g/L)
Nifoss 2500: make-up	200
Nifoss 2500: base	60
DI water	740

**Table 4 materials-15-06806-t004:** Thickness, microhardness, and surface roughness of each electroless coating.

Experiment Nr.	Powder Concentration (g/L)		Coating Thickness (µm)	Microhardness (HV)	Average Surface Roughness (Ra) (µm)
	Al_2_O_3_	MoS_2_			
1	0	0	8.23	244.10	0.45
2	2	0.5	4.49	296.85	0.64
3	2	1.0	1.22	256.62	0.66
4	2	1.5	2.02	241.69	0.59
5	2	2.0	4.20	222.95	0.55
6	3	0.5	6.16	305.09	0.78
7	3	1.0	1.63	266.55	0.85
8	3	1.5	2.84	254.94	0.80
9	3	2.0	4.75	227.73	0.73
10	4	0.5	3.91	315.83	0.69
11	4	1.0	1.11	277.55	0.75
12	4	1.5	1.65	268.64	0.73
13	4	2.0	3.89	254.08	0.69

## Data Availability

Not applicable.

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
