# Peer review of "Electroless Ni-P-MoS2-Al2O3 Composite Coating with Hard and Self-Lubricating Properties"

_materials, 2022, doi:10.3390/ma15196806_

Round 1

Reviewer 1 Report

Give the full form of the abbreviations at their first appearance.

Bulk citation at a single statement should be avoided “The performance evaluation of Ni-P coated surfaces is con- 51 ducted in context to the factors mentioned above in several studies [8,9,11,13,17,18,22]

Consider some recent literatue on NiP Coatings.

Chintada, Vinod Babu, Ramji Koona, and M. V. A. Raju Bahubalendruni. "State of Art Review on Nickel-Based Electroless Coatings and Materials." Journal of Bio-and Tribo-Corrosion 7.4 (2021): 1-14.

Shacham-Diamand, Y., et al. "30 years of electroless plating for semiconductor and polymer micro-systems." Microelectronic Engineering 132 (2015): 35-45.

Please improve the quality of figure 2, figure 3 & 4, 6

Overall, the manuscript is well written and substantiated with experimental results.

All the best.

Author Response

We thank the reviewer for the comments. The manuscript has been revised based on the comments in a satisfactory manner. We hope the manuscript will be suitable for publication in the present form.

Comment 01: Give the full form of the abbreviations at their first appearance.

Response: The full form of the abbreviations has been added to the revised manuscript at their first appearance.

Comment 02: Bulk citation at a single statement should be avoided “The performance evaluation of Ni-P coated surfaces is conducted in context to the factors mentioned above in several studies [8,9,11,13,17,18,22]

Response: The bulk citation has been removed as per suggestion.

Comment 03: Consider some recent literatue on NiP Coatings.

Chintada, Vinod Babu, Ramji Koona, and M. V. A. Raju Bahubalendruni. "State of Art Review on Nickel-Based Electroless Coatings and Materials." Journal of Bio-and Tribo-Corrosion 7.4 (2021): 1-14.

Shacham-Diamand, Y., et al. "30 years of electroless plating for semiconductor and polymer micro-systems." Microelectronic Engineering 132 (2015): 35-45.

Response: New references have been added as per suggestion.

Comment 04: Please improve the quality of figure 2, figure 3 & 4, 6

Response: All the figures have been modified as per the suggestion.

Author Response

We thank the reviewer for the comments. The manuscript has been revised based on the comments in a satisfactory manner. We hope the manuscript will be suitable for publication in the present form.

Comment 01: No novelty in this work

Response: The present study focusses on the incorporation of multiple attributes i.e., hardness, self-lubrication, hydrophobicity, and wear-resistant properties on a single material (aerospace grade Al-alloy: Al7075) through electroless deposition route. An extensive analysis is done on the surface morphology, topography, hardness, composition, roughness, wettability, and coating thickness for various powder ratios (MoS2 and Al2O3) in the electroless bath.

Comment 02: The manuscript requires English language editing to improve the quality of the presentation. Due to many grammar and spelling errors, in many parts, the manuscript was hard to follow.

Response: The grammatical errors and the English correction has been done using Grammarly software (premium) version to rectify the mistakes.

Comment 03: Regarding to “MoS2-Al2O3 powder” must be corrected to “MoS2-Al2O3 reinforcements” in all the text.

Response: The suggestion has been incorporated in the revised manuscript.

Comment 04: Abstract must be re-write and mention all the results of the study

Response: The abstract has been modified as per the suggestion.

Comment 05: Regarding to “the corrosive resistance” should be corrected to “the corrosion resistance” in all the text.

Response: The word “corrosive resistance” has been changed to “corrosion resistance” at all places throughout the revised manuscript.

Comment 06: New references for the basic of electroless technique on the different substrate must be added like:

- Hamid, Z.A., El-Adly, R.A. Mechanism of Nickel-Phosphorus-Aluminum Oxide Composite

Coatings by Electroless Process, Plating and Surface Finishing, 1999, 86(5), pp. 136–139

- Hamid, Z.A., Mechanism of electroless deposition of Ni-W-P alloys by adding surfactants,

Surface and Interface Analysis, 2003, 35(6), pp. 496–501

Response: New references have been added as per suggestion.

Comment 07: In the experimental section: Fig. 1 is very poor and must be improved

Response: Figure 1 has been improved as per suggestion.

Comment 08: In the Table 2 “Fe3Cl” must be corrected to “FeCl3”.

Response: The correction has been made as per suggestion.

Comment 09: The chemical composition of electroless bath must be written in detail, what do you mean by: Nifoss 2500: make-up 200 and Nifoss: 2500:base?

Response: Nifoss is a commercially available electroless plating bath procured from CMP Industries, Kolkata, India for using in the study.

Comment 10: The particle size of the reinforcement must be mentioned in the text.

Response: Al2O3: 600 nm and MoS2: 7 μm was used in the present context of study.

Comment 11: Regarding to “stirring limit was fixed to 600 rpm”, from my experience this rpm is very high and may be led to weep the particles from the surface and consequently decrease the codeposition of the particles in the matrix so it’s very important to study the effect of stirring rate

Response: The stirring speed was decided as per the previous studies (10.1007/s12666-019-01677-1). The authors accept that there might be weeping of the particles from the coated region, and thus, will work on the suggestion in the near future.

Comment 12: The authors mentioned that “In Fig. 3(h) that when MoS2 powder increases from 0.5 g/L to 1g/L at a constant value of Al2O3, coating thickness decreases drastically, but further addition of MoS2 powder to EL-bath, i.e., at 1.5 g/L and 2 g/L, coating thickness continue to increase” explain why

Response: The reason for the above phenomenon is as follows:

“There is a homogeneous deposition of Al2O3 and MoS2 powder in the Ni-matrix. This may be the reason behind the maximum coating thickness. Moreover, Al2O3’s abrasive nature creates hindrance whilst the deposition. A larger Al2O3 (4 g/L) concentration may create random movement in the EL-bath, suppressing the deposition process [32]. Fig. 4(g) depicts the morphology of surface coating with Al2O3 4 g/L and MoS2 1 g/L powder content. It clearly displays the heterogeneity in the powder distribution, which may lead to poor deposition and hence the least coating thickness.”

This has been incorporated in the revised manuscript.

Comment 13: Where is the value of the roughness (Ra) of the substate before the coating process?

Response: The surface roughness of aluminum substrate prior to the deposition process was found to be 0.36 µm.

Comment 14: The authors must be compared these results with the other related works?

Response: The authors have now compared the results with other related works as per suggestion.

Comment 15: In the Scratch test: the authors mentioned that "the COF without the powders was 0.13, for samples coated using powders was 0.12" the o.12 value does not improve as shown in Fig. 10c

Response: The authors have estimated the average coefficient of friction for the samples, so there is a low difference in the COF values.

Reviewer 3 Report

This paper studies an electroplating coating with multiple properties. The hardness, self-lubricating, hydrophobic and wear-resisting properties are studied extensively, and the research content is more substantial. However, there are some unsolved experimental phenomena and details to be dealt with in this paper. Some are as follow:

1. A reasonable explanation should be given as to why the coating thickness increases first and then decreases with the increase of MoS2.

2. In Figure 9, (a) and (b) should be labeled more clearly, and why the peaks of MoS2 profess the improvement in the lubricity properties of the sample should be explained more detail.

3. Although the addition of powders enhances the wear-resistant properties, the mechanism is not explained

4. In Figure 2, Al2O3 exists in many phases and should be accurately marked in XRD.

Author Response

We thank the reviewer for the comments. The manuscript has been revised based on the comments in a satisfactory manner. We hope the manuscript will be suitable for publication in the present form.

Comment 01: A reasonable explanation should be given as to why the coating thickness increases first and then decreases with the increase of MoS2.

Response: The reason for the above phenomenon is as follows:

“There is a homogeneous deposition of Al2O3 and MoS2 powder in the Ni-matrix. This may be the reason behind the maximum coating thickness. Moreover, Al2O3’s abrasive nature creates hindrance whilst the deposition. A larger Al2O3 (4 g/L) concentration may create random movement in the EL-bath, suppressing the deposition process [32]. Fig. 4(g) depicts the morphology of surface coating with Al2O3 4 g/L and MoS2 1 g/L powder content. It clearly displays the heterogeneity in the powder distribution, which may lead to poor deposition and hence the least coating thickness.”

This has been incorporated in the revised manuscript.

Comment 02: In Figure 9, (a) and (b) should be labeled more clearly, and why the peaks of MoS2 profess the improvement in the lubricity properties of the sample should be explained more detail.

Response: Figure 9 has been edited as per suggestion. The possible improvement in the lubricity properties due to the presence of MoS2 is explained satisfactorily in the revised manuscript.

Comment 03: Although the addition of powders enhances the wear-resistant properties, the mechanism is not explained

Response: The explanation for improved wear resistant properties have been incorporated in the revised manuscript.

Comment 04: In Figure 2, Al2O3 exists in many phases and should be accurately marked in XRD.

Response: The unidentified peaks have been added in the XRD plot as per suggestion.

Round 2

Reviewer 2 Report

The authors have adequately addressed the concerns raised by reviewers.

Author Response

Thank you for the comments.

Reviewer 3 Report

For the revised version, the author must address the phase structure of Al2O3 in Fig.2. Furthermore, the relevant analysis is little. The English should be polished. The author should further check the references, which should be suitable for citing.

Author Response

Thanks for the comments and constructive criticism. We have modified the manuscript according. Please find the response below.

Comment 01: For the revised version, the author must address the phase structure of Al2O3 in Fig.2. Furthermore, the relevant analysis is little.

Response: The Al2O3 powders were present as α-Al2O3 with hexagonal structure acquired from reference pattern 00-026-0031. Relevant information has been included in the revised version of the manuscript.

Comment 02: The English should be polished.

Response: The revised manuscript has been checked by English professional software (Grammarly) to remove grammatical and language errors.

Comment 03: The author should further check the references, which should be suitable for citing.

Response: New references have been added as per the suggestion.